# Techniques and Materials for Optical Fiber Sensors Sealing in Dynamic Environments with High Pressure and High Temperature

**DOI:** 10.3390/s21196531

**Published:** 2021-09-30

**Authors:** Joao Batista Rosolem, Rivael Strobel Penze, Claudio Floridia, Rodrigo Peres, Deleon Vasconcelos, Marcelo Agra Ramos Junior

**Affiliations:** 1CPQD Research and Development Center in Telecommunications, Campinas 13086.902, Brazil; rpenze@cpqd.com.br (R.S.P.); floridia@cpqd.com.br (C.F.); rperes@cpqd.com.br (R.P.); 2Centrais Elétricas da Paraíba, João Pessoa 58071.973, Brazil; deleon.vasconcelos@utepasa.com.br (D.V.); marcelo.agra@utepasa.com.br (M.A.R.J.)

**Keywords:** harsh environments, high pressure, high temperature, materials, optical fibers, optical sensors, sealing

## Abstract

We detail a study of the techniques and sealing materials for optical fiber sensors used in dynamic environments with high pressure (>300 bar) and high temperature (>300 °C). The sealing techniques and materials are the key for the robustness of sensors in harsh dynamic environments, such as large combustion engines. The sealing materials and techniques studied in this work are high-temperature epoxies, metallic polymer, metallic solders, glass solder, cement, brazing and electroless nickel plating. Because obtaining high temperature simultaneously with high pressure is very difficult in the same chamber in the laboratory, we developed a new and simple method to test sealed fibers in these conditions in the laboratory. In addition, some sensors using the materials tested in the laboratory were also field tested in real thermoelectric combustion engines. The study also discusses the methods of fabrication and the cost−benefit ratio of each method.

## 1. Introduction

Optical fiber sensors have been gradually studied and tested in many engineering applications. Sensing in harsh environments, such as aerospace, marine, oil and gas, and thermal electrical energy generation is a study area of great importance where optical fiber sensors have advantages compared with electric/electronic sensors due to their good characteristics such as robustness, flexibility, cost, and availability. Some previous studies have reported the use of optical fiber sensors in harsh environments [1,2,3].

The particular harsh environments of interest are the ones with dynamic high temperature and high pressure, such as the combustion chambers of gas and diesel engines, jet turbines, and rocket boosters. The fixing method of the optical fibers of the sensors in the packaging is the key factor for sensor performance in these environments.

We detailed in this work a study of techniques and sealing materials for optical fiber sensors used in dynamic environments with high pressure (>300 bar) and high temperature (>300 °C). The research started with a project to develop internal combustion pressure sensors for thermoelectric engines. In thermoelectric applications, it is relevant to maximize the engine operation to substantially reduce nitrogen oxides (NOx) and soot emissions and improve the engine’s fuel economy. The development of the internal combustion pressure sensor for engines is a relevant issue to obtain a closed-loop control of the mass fraction burned for many engines. The standard piezoelectric sensors used to measure the pressure of the combustion chamber currently are not durable in high temperatures when they are used continuously. Thus, this application needs a robust and trusty pressure sensor. It is important to comment that in a thermoelectric power plant there are dozens of engines, each one with dozens of cylinders. In the research we conducted, the ultimate goal was to install one sensor internally on each engine cylinder nozzle, FBG pressure sensors are ideal to be connected in a sensors network, in such a way, using a single interrogator. In addition, FBG in silica fiber is a technology available for the range of temperatures and pressures required in this application. Special packaging materials and sealing techniques for FBG must be considered to obtain a long life pressure sensor. The research team call this region (cylinder nozzle) “planet Venus” because an extremely harsh environment similar to that on Venus surrounds the optical sensor at the recommended point to measure pressure (inside the combustion chamber of the engine).

The techniques and sealing materials studied in this work are high-temperature epoxies, metallic polymer, metallic solders, glass solder, cement, brazing and electroless nickel plating. To the best of our knowledge, this is the first demonstration of optical sensor sealing using metallic polymer and glass solder.

The sealing techniques studied in this work were applied to FBG pressure sensors, but they can be also applied for single-end sensors, such as Fabry−Perot or other reflective types. Because dynamic high temperature simultaneously with dynamic high pressure is very difficult to obtain and control in the same chamber in a laboratory, we developed a new and simple method to test the sealing materials fixed in the fibers by having these conditions simultaneously in the laboratory. In addition, some FBG sensors using some of the sealing materials tested in the laboratory were also tested in real thermoelectric combustion engines. Moreover, the study discusses the methods of fabrication and the cost−benefit ratio of each method. However, it is not the focus of this work to evaluate in detail the sensor performance in terms of response to pressure or temperature measurements, but the focus is to investigate the robustness of the sealing materials and techniques employed in the sensors. Many details of sensor performance can be accessed in [4].

The work was organized as follows: in Section 2, we presented the optical fiber sealing scheme and the materials, characteristics, and sealing techniques investigated in this work. In Section 3, we detailed three experimental tests that we used to validate the sealing materials and techniques in FBG pressure optical sensors. Section 4 presented the results and discussion of the three tests used to evaluate the sealing techniques. Finally, in Section 5 we presented the conclusions.

## 2. Methods and Materials

### 2.1. The Optical Fiber Sensor Sealing Scheme

The sealing material must be robust to the dynamic actuation of high pressure (>300 bar) and the high temperatures (>300 °C). As we commented in the introduction, some examples of dynamic environments of high pressure and temperature are the combustion chambers of gas and diesel engines, jet turbines, and rocket boosters.

The optical fiber sealing scheme is shown in Figure 1. The optical fiber sensor (single-ended type) is inserted into the high temperature and pressure environment through a hole inside the pressure connector, which is fixed to the chassis wall that encases this environment. The optical fiber is sealed in the hole by applying the sealing material inside this hole and in the top of the connector. As we commented before, our ultimate goal was to install one sensor internally on each engine cylinder nozzle.

To seal the fiber in this environment the sealing material must be properly chosen for use in high temperature and in high pressure. Some material parameters that should be analyzed are maximum service temperature, coefficient of thermal expansion (α), melting temperature point, Young’s modulus, and compressive strength.

A classical approach is to make the matching of the coefficient of thermal expansion of the materials or, in a no-matching case, choose a material with a coefficient of thermal expansion higher than the glass to have glass compression [5]. However, in high temperatures, other effects can appear that compromise the final quality of the sealing [6]. Weight loss in high temperatures for sealing materials directly affects the efficiency of the seal. In addition, considering all technical aspects, the sealing process must be cost-effective.

### 2.2. Materials and Techniques

There are some techniques and materials that have been investigated for optical fiber sealing but some of them are limited to low temperatures applications (<150 °C). In this work, we studied some materials applied for temperatures higher than 300 °C.

#### 2.2.1. Epoxy

Epoxies are the most common, cheap, and easy materials to work as sealing materials. There are many options for use in high temperatures and many of them have been subjected to rigorous performance tests in aerospace applications [7], in electronics applications [8], and in optical sensing applications [9,10].

The viscoelastic properties of epoxy alter in high temperatures. As temperature increases, a significant amount of the flexural and compressive strength of epoxy decreases. When the temperature increases epoxy reaches the heat distortion temperature (HDT), and it begins to deform. The HDT of an epoxy correlates to its glass transition temperature. The continued increase in temperature leads to more ductile behavior. The temperature increase also leads to a loss of load-bearing capacity and stiffness. In addition to temperature, moisture and ultraviolet radiation influence the breaking down of the epoxy matrix. The high-temperature epoxies are specially formulated to withstand temperatures higher than 250 °C.

Figure 2 shows some samples of optical fiber ferrules filled with the epoxy H74-100 (from Epo-Tek) [11]. We used this epoxy in the performance tests described in the next section. The hole of the ferrules containing an optical fiber was filled with epoxy. This epoxy was also inserted into the top of a ceramic ferrule. Figure 2a shows some samples of assembled ferrules after the epoxy curing process. Figure 2b shows a type of pressure connector (pressure adapter type 5/32” × 1/8 NPT male) where the ferrules were inserted.

In the preparation, the two parts of the epoxy need to be mixed carefully. The parts where the epoxy will be used must be cleaned previously. Finally, after the epoxy is inserted in the ferrule hole, the assembled parts must be heated for the time and at the temperature specified by the supplier.

In general, the process for use of high-temperature epoxies is not complex and the cost for the specific epoxy we tested is around USD 1.02/g.

#### 2.2.2. Polymer Composite

A polymer composite or polymer blend is a mixture of two or more materials with the polymer that has been blended to create a new material with different physical properties. This type of material has been used, for example, to fix corroded metallic pipes used in the oil industry [12,13]. This material can have superior performance as compared with high-temperature epoxies. One type of polymer composite that uses metals in its composition is a metallic polymer. This metallic polymer is suitable for use in high temperatures.

It is possible to apply metallic polymers to transformer radiators, oil sump, engine blocks, pipelines, and fuel tanks, even if they are in operation. They can withstand temperatures ranging from −40 to 500 °C and pressure up to 300 bar. That is, metallic polymers are materials with excellent strength and durability. In addition, another characteristic that can be found in polymers is that they can be molded, rectified, and machined, thus becoming highly versatile materials.

Figure 3a shows the hole of an adapted pressure connector (3/8-24 UNF (M10 × 1.0) filled with the metallic polymer VP 10–500 (from MultiMetall) [14]. Figure 3b shows the pressure connector using the polymer as a sealing technique.

In the preparation of the polymer, its two parts need to be mixed carefully. The parts where the polymer will be used must be cleaned previously. Finally, after the polymer is inserted in the connector hole the assembled parts must be heated for the time and at the temperature specified by the supplier.

In general, the process for using high-temperature metallic polymers is not complex and the cost for the specific metallic polymer we tested is around USD 1.4/g.

#### 2.2.3. Cement

High-temperature cement is ceramic cement based on different inorganic materials such as silica, alumina, sodium silicates, magnesium oxide, etc. These types of cement have been used for many years for general high-temperature research, electronics, metallurgical, nuclear, and industrial applications. Ceramic cement is commonly applied for strain gauge applications [15,16] and it has been reported in previous applications for FBG sensors [17]. Some of the characteristics of ceramic cement are its resistance to temperatures over 700 °C, oil, solvents, and most acids; it is heat conductive, thermal shock resistant and is an electrical insulator, it adheres to metals, ceramics, glass, porcelain, and most other surfaces and has excellent mechanical bonding characteristics. The cements exhibit excellent bonding characteristics and are used on metals or other materials which have a high coefficient of thermal expansion.

Figure 4a shows the hole of an adapted pressure connector (3/8-24 UNF (M10 × 1.0) filled with the ceramic cement OB-700 (from Omega) [18]. Figure 4b shows the pressure connector using the polymer as a sealing technique.

The cement is prepared with water and needs to be mixed carefully. The parts where the cement will be used must be screwed and cleaned previously. Finally, after the cement is inserted in the connector hole the assembled parts must be heated for the time and at the temperature specified by the supplier.

In general, the process for using the high-temperature cement is not complex and the cost for the specific cement we tested is around USD 0.32/g.

#### 2.2.4. Glass Solder

Glass solder preforms have been used for hermetic sealing of optical fibers in optoelectronic and optical packaging [19]. The preform adheres well to a wide variety of metals, glasses, ceramics, and semiconductor materials. Applications include sealing glass of optical fibers in a metal package without fiber metallization and they replace metal solders. There is a wide variety of shapes and sizes of glass solder preforms. Different models are available, depending on the melting point. After the soldering processing in optoelectronic and optical packaging, the assemblies sealed with glass solder can meet the stringent humidity resistance, hermeticity and strength demanded of component packaging.

For fixing the fiber in the pressure sensor, we chose the model DM2760PF (RP30-11-20 outside diameter = 0.762 mm, internal diameter = 0.279 mm, and thickness = 0.508 from Diemat) which had the following characteristics [20]: melting point between 380 to 450 °C, glass transition temperature (Tg) = 260 °C and coefficient of thermal expansion = 6.9 ppm/°C.

There are several methods for melting the glass preforms [19]. Once the glass solder preform melting temperature is higher than metallic solder, we adapted a temperature-controlled 150 W soldering iron (Figure 5a) to melt the glass preforms (Figure 5b) into the optical fiber in the top of a pressure connector type 5/32” × 1/8 NPT male. The Appendix A shows the melting of glass preforms.

The process of sealing the fiber is more complex using this technique than the previous ones. To observe and control the melting process, one iron solder, one microscope, and one thermopile coupled to a multimeter are necessary. The fiber to the glass interface is very fragile and needs coating protection before removal from the iron solder platform. The cost of the specific glass solder we tested is around USD 10.16/g.

#### 2.2.5. Metallic Solder

Metallic solders are an option to search for high temperatures in the sealing process. To use metallic solders, the optical fiber end should be previously metalized with gold to have a hard fixing with the metallic solder. Of course, the gold metallization on the fiber also protects it from glass erosion in harsh environments. Two techniques involving metallic solders have been used to seal the optical fiber in pressure connectors: soldering and brazing [21,22,23]. Soldering is a process in which two or more items are joined together by melting and putting a filler metal (solder) into the joint, the filler metal having a lower melting point than the adjoining metal. In brazing, the workpiece metal also does not melt, but the filler metal melts at a higher temperature than in soldering. There are metallic solders for a wideband range of temperature operations. We chose two metallic solders to investigate the sealing process using soldering.

The first solder is the high melting point (HMP) Sn05Pb93.5Ag1.5 [24]. This solder presets a melting point between 296 to 301 °C. However, the maximum safe service temperature to the solder alloy subjected to stress is about 40 °C below the solidus melting temperature. HMP alloy can therefore be relied upon in service up to about 255 °C.

Figure 6a shows the process to solder the gold metalized fiber that contains the FBG in the pressure connector pressure connector type 5/32” × 1/8 NPT male and Figure 6b shows the result of the soldering process. The soldering process uses a common controlled temperature 50 W soldering iron. Care should be exercised to avoid unnecessarily high tip temperatures for an excessive period. A high-temperature solder tip will increase the flux spitting and it may produce some residue darkening. In this process, the fixative material (metallic solder) melts and joins the fiber, the pressure connector body, and the nickel ferrule. This sealing technique is very simple needing just only a simple iron solder as a tool but the fibers need to be gold metalized. The cost for the specific HMP solder we tested is around USD 0.076/g.

The second solder is the SSQ-6 (from Muggy Weld) [25] that is a high-strength silver solder paste with a high silver content of around 56%.

This paste can be used with a variety of metals at over 565 °C and tensile strength of 85,000 psi. It can be used on cast iron, as well as for bonding stainless steel, brass, copper, bronze, mild steel, tool steel, carbide, carbon steel, and chrome-moly individually or with other metals. It can be soldered using propane, MAPP (methylacetylene-propadiene propane) gas, or oxyacetylene.

Figure 7a shows the process to solder the gold metalized fiber that contains the FBG in a pressure connector using the SSQ-6 solder. Figure 7b shows the result of the soldering process and Figure 7c the entire pressure connector (3/8-24 UNF (M10 × 1.0). This soldering process used an oxyacetylene flame to melt the solder in a quick time. This process depends on the ability of the welder and it is very dangerous to damage the optical fiber due to the high temperatures (>1000 °C) reached by the oxyacetylene flame. Again, care should be exercised to avoid unnecessarily high tip temperatures for excessive times. In this process, the fixing material (metallic solder) melts and joins the fiber, at the pressure connector body.

Figure 6a shows the process to solder the gold metalized fiber that contains the FBG in the pressure connector and Figure 6b shows the result of the soldering process. The soldering process uses a common controlled temperature 50 W soldering iron. Care should be exercised to avoid unnecessarily high tip temperatures for an excessive period. A high-temperature solder tip will increase the flux spitting and it may produce some residue darkening. In this process, the fixative material (metallic solder) melts and joins the fiber, the pressure connector body, and the nickel ferrule.

This sealing technique is very simple needing only a simple iron solder as a tool but the fibers need to be gold metalized. The cost for the specific HMP solder we tested is around USD 0.076/g.

The last process using solder that we investigated for sealing the optical fiber in the pressure connector is brazing. In the brazing sealing process of the pressure connector, the gold metalized fiber, and the alloy were carried out by vacuum at a temperature of 900 °C. Typical alloys used in this process are Cusil, Palcusil 10, Cusin 1 ABA e Ticusil [26]. Figure 8a shows the blazing material inside the pressure connector and Figure 8b shows the fiber in the outside part of the pressure connector.

Figure 6a shows the process to solder the gold metalized fiber that contains the FBG in the pressure connector (3/8-24 UNF (M10 × 1.0) and Figure 6b shows the result of the soldering process. The soldering process uses a common controlled temperature 50 W soldering iron. Care should be exercised to avoid unnecessarily high tip temperatures for an excessive period. A high-temperature solder tip will increase the flux spitting and it may produce some residue darkening. In this process, the fixing material (metallic solder) melts and joins the fiber, the pressure connector body, and the nickel ferrule. This sealing technique is very simple needing only a simple iron solder as a tool but the fibers need to be gold metalized. The cost for the specific HMP solder we tested is around USD 0.076/g.

#### 2.2.6. Nickel-Plated Sealing

Electroless plating is a method where a metal film is deposited on a base material through catalyzed chemical reduction without external current of solution-phase metal ions at the substrate surface. Electroless plating has been used widely for automotive, hardware, and electronic applications and in optical fiber sensors [27,28].

We performed previous studies of electroless nickel plating in nickel ferrules. Figure 9a shows the chemical nickel/phosphorous bath (NI-422 supplied from Enthone) during a fiber/ferrule metallization process. Figure 9b shows the fiber ferrule sealed in the nickel ferrule after the metallization process. The thickness of the electroplated nickel is related directly to the electroplating time. To grow a nickel layer of around 20 μm, the required bath time is 16 h at a temperature of 85 °C. The nickel-plated sealing was done in sensors using modified pressure connectors type 5/32” × 1/8 NPT male.

## 3. Experimental Tests and Set-Ups

As we commented before, it is not the focus of this work to evaluate the sensor performance in terms of pressure or temperature measurements in detail, but the focus is to investigate the robustness of the sealing materials and techniques employed on the sensors. The sensor performance can be accessed, for example, in [4] or [29].

### 3.1. Test 1—Endurance

Because dynamic high temperature simultaneously with dynamic high pressure is very difficult to obtain and control in the same chamber in a laboratory, we developed a new and simple method to test the sealing materials fixed in the fibers simulating these conditions in the laboratory. In the endurance test, the optical fibers were sealed at one end of the nickel ferrules using the materials and techniques described in Section 2. These samples were submitted to a high temperature and a pull that simulated the pressure on the fiber.

Three ferrule samples of each sealing material were placed inside a 150 W soldering iron tip, which heated the ferrule over 600 °C. Figure 10a shows the schematic of the setup utilized, Figure 10b shows a photo of the real setup and Figure 10c shows a zoom of the ferrule with the fiber sealed in the heating region of the soldering iron. In this test, the fiber was pulled using a mass of 38 g, simulating in this way the pressure of 300-bar acting in the fiber sealing, simultaneously with the temperature increasing. When the maximum temperature of the sealing operation is reached, the fiber comes off the seal and falls out. The temperature was measured using one class K thermocouple. The thermocouple was fixed very close to the hole containing the nickel ferrule.

### 3.2. Test 2—Leakage

In the leakage test, the pressure sensors packaged with optical fibers sealed using the materials and methods shown in Section 2, were tested at room temperature using a static pressure generator WIKA CPP-700-H that generates pressure (Figure 11)). Each pressure sensor was characterized under different pressure levels up to 300 bar and kept at 300-bar for 1 h to observe that no leak was present in the sensor due to the sealing process.

### 3.3. Test 3–Field Test in a Thermoelectric Engine

The field tests were performed in Centrais Elétricas da Paraíba (EPASA), which is a thermoelectric power plant. This thermoelectric power plant has an installed power of 340 MW, obtained from 40 engines model MAN 3240. The angular speed of each motor is 720 rpm, and heavy fuel oil (OCB1) is used in the combustion engines. Each engine has 18 cylinders and uses a mechanical injection pump to control the fuel oil injection, reducing the possibility of adjustments in the injected fuel volume and the same proportion, limiting the management of the engines.

The thermoelectric power plant is a harsh environment. The internal temperature in the machine room can reach 55 °C and extraordinary events such as the tests of a new type of sensors are limited in terms of time and number of sensors. On the other hand, the data collected in the field are useful to improve the design of the sensors. The tests were performed during short periods in four seasons.

The objective of the field test was to verify the performance of the sensors in terms of robustness in an environment with variable temperature and pressure. In the field tests, the sensors were connected to a pressure monitoring point available for each engine cylinder.

Sensors using the four sealing material types were tested in the field test: epoxy, metallic polymer, metallic solder, and nickel-plating. Figure 12a,b show a sensor sealed with polymer connected to an engine pressure monitoring point.

It is interesting to comment that normally this monitoring point is used for occasional pressure measurements in the cylinders. This periodic evaluation is essential to monitor the status of the engine and its individual cylinders. The measurement of data at this point is done using a standard sensor system (e.g., HLV 4.0 from Kistler). The time the standard sensor remains connected to the monitoring point is limited to a few minutes to avoid damage.

## 4. Results and Discussion

The tests described in Section 3 were performed for verification of the sealing efficiency of each material. Three samples were prepared for Test 1 using all materials and techniques described in Section 2, except for the solder SSQ-6 and brazing technique. The high temperature tools damaged all the samples prepared with these techniques. In Table 1, we can observe in the temperature endurance test (Test 1) that the temperature ranged from 282 to 622 °C, depending on the sealing material. In addition, the high-temperature endurance obtained from the cement material did not guarantee good performance in the leakage test (Test 2).

Figure 13 shows the results of the leakage test for the sensors based on polymer, nickel-plating, glass solder, metallic solder and epoxy. The sensitivity of the sensor based on polymer was lower than the other materials because a thick layer of gold covered the FBG used in this sensor. We used an FBG interrogator model si155 Hyperion from Micron Optics to characterize the sensors. Eventually the sealing process caused micro bends on the fiber surface and increased the loss. However, according to the pressure characterization shown in Figure 13, the sealing process did not affect the pressure sensitivity of the sensors.

Four sealing material types were tested in the field test: epoxy, nickel-plating, metallic solder, and metallic polymer. In the packaging of the sensors using these sealing materials were used two types of pressure connectors and fiber protections as we can see in Figure 2, Figure 3, Figure 6 and Figure 9. However, in all cases the optical fiber sensor was inserted in the high temperature and pressure environment through a hole inside the pressure connector, which was fixed in the chassis wall that encases this environment.

Figure 14 shows the temporal response of the pressure sensor using nickel-plating sealing at the exact instant that the sealing degradation started (90 min after test initialization) at a pressure sensor temperature of 345 °C. This temperature was obtained in a previous testing of the engine with different loads [29]. The temperature of sealing degradation was in accordance with the previous results obtained in Test 1 for the nickel-plating sealing technique.

Autopsies performed on the sensors based on epoxy, metal solder and nickel plating after the field tests revealed that the sealing material was completely removed from the connector hole. On the other hand, for the metallic polymer sensors the sealing material remained intact in the pressure connector hole.

Table 2 illustrates a resume of the analysis for complexity and the cost for each sealing material/technique described before. Each material can attain a maximum temperature of operation, but the complexity and the cost of the technique and the materials do not always compensate for its use.

We consider metallic polymer as the better material for optical fiber sealing in our application, considering the three tests that we applied to the sealing materials, such as, endurance, leakage and field test, and the cost-benefit analysis considering the complexity of sealing the sensors. For all these criteria, the performance of the metallic polymer was better for the temperature from 300 to 500 °C and pressure from 0 to 300 bar. In addition, we believe that the sealing technique can be improved if the sealing layer could add more nickel mass. According to the studies demonstrated in [28], this type of material can resist to 900 °C.

For other temperature ranges and for other types of fibers, new techniques can be used. For example, in [30], a sapphire pressure-sensitive chip (temperature operation > 1000 °C) is packaged with a high-temperature alloy material by an elastic sealing technique. In [31], a low temperature application is described for monitoring the structural integrity of adhesively bonded joints by integrating a polymer optical fiber (POF) into the adhesive layer using a two-component polyurethane adhesive 3 M Scotch-Weld DP 609.

## 5. Conclusions

We detailed a study of techniques and sealing materials for optical fiber sensors used in dynamic environments with high pressures (300 bar) and high temperatures (>300 °C). Some sealing materials (high-temperature epoxies, metallic and glass solders, cement, brazing and electroless nickel plating) and associated techniques were described, tested, and evaluated.

Our research was directed towards applications with internal combustion pressure measurement for thermoelectric engines. In thermoelectric applications, it is relevant to maximize the engine operation to substantially reduce nitrogen oxides (NOx) and soot emissions and improve engine fuel economy. The development of the internal combustion pressure sensor for engines is a relevant issue for obtaining a closed-loop control of the mass fraction burned for many engines. The standard piezoelectric sensors used to measure the pressure of the combustion chamber currently are not durable in high temperatures when they are used continuously. Thus, this application needs a robust and trusty pressure sensor. It is important to comment that in a thermoelectric power plant there are dozens of engines, each one with dozens of cylinders. In the research we conducted, the ultimate goal was to install one sensor internally on each engine cylinder nozzle. In such a way, FBG pressure sensors are ideal to be connected in a sensor network, using a single interrogator. In addition, FBG in silica fiber is a technology available for the range of temperatures and pressures required in this application. Special packaging materials and sealing techniques for FBG must be considered to obtain a long life for the pressure sensor.

The metallic polymer was the better material for optical fiber sealing in our application, considering the three tests that we applied to the sealing materials: endurance, leakage and field test, and the cost−benefit analysis considering the complexity of sealing the sensors. In all these criteria, the performance of the metallic polymer was better for the temperature from 300 to 500 °C and pressure from 0 to 300 bar.

Other materials can also be used in different temperature or pressure ranges but the complexity and cost associated must be evaluated.

## Figures and Tables

**Figure 1 sensors-21-06531-f001:**
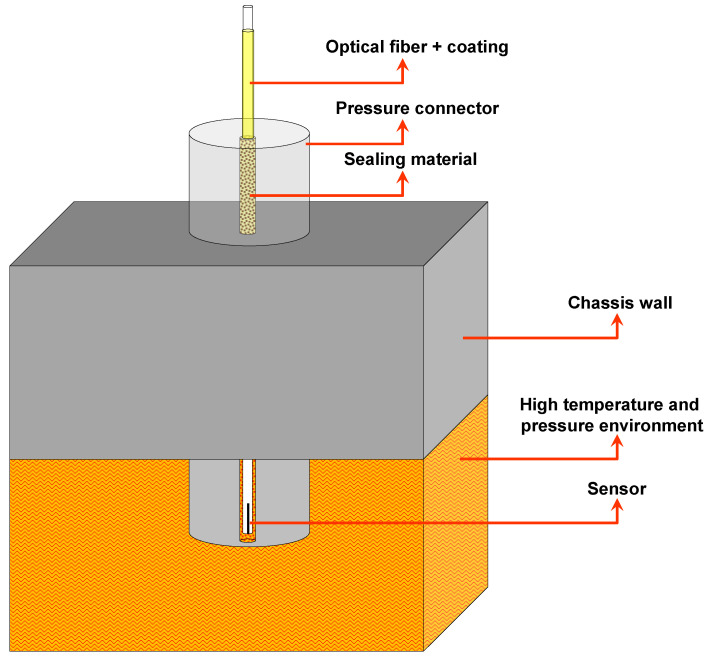
The sealing scheme for the optical fiber sensors.

**Figure 2 sensors-21-06531-f002:**
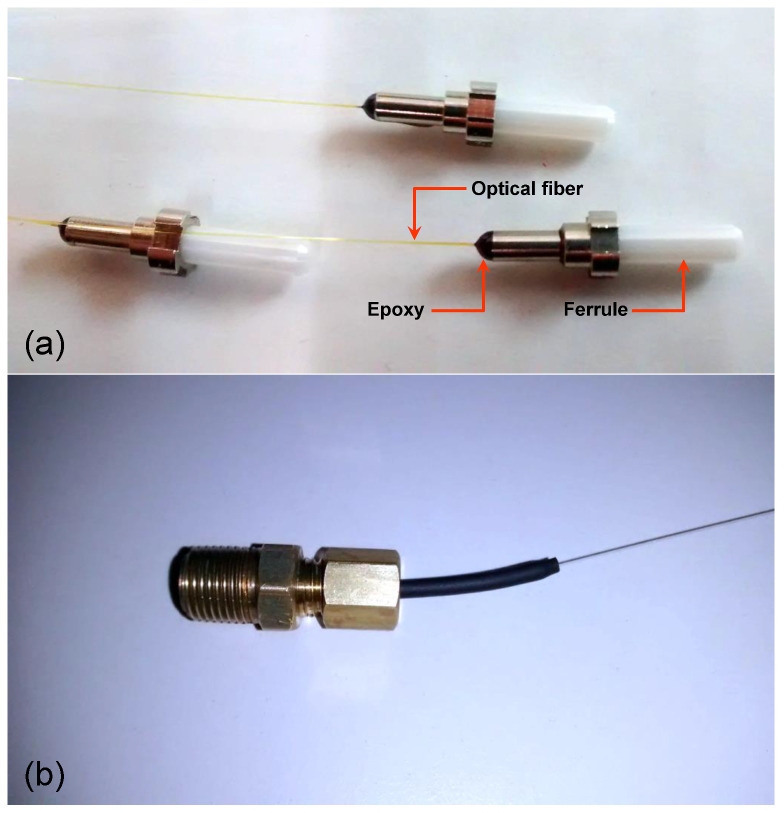
(**a**) Optical fiber ferrules assembled with optical fiber and the high temperature epoxy and (**b**) a pressure connector using the sealed ferrules.

**Figure 3 sensors-21-06531-f003:**
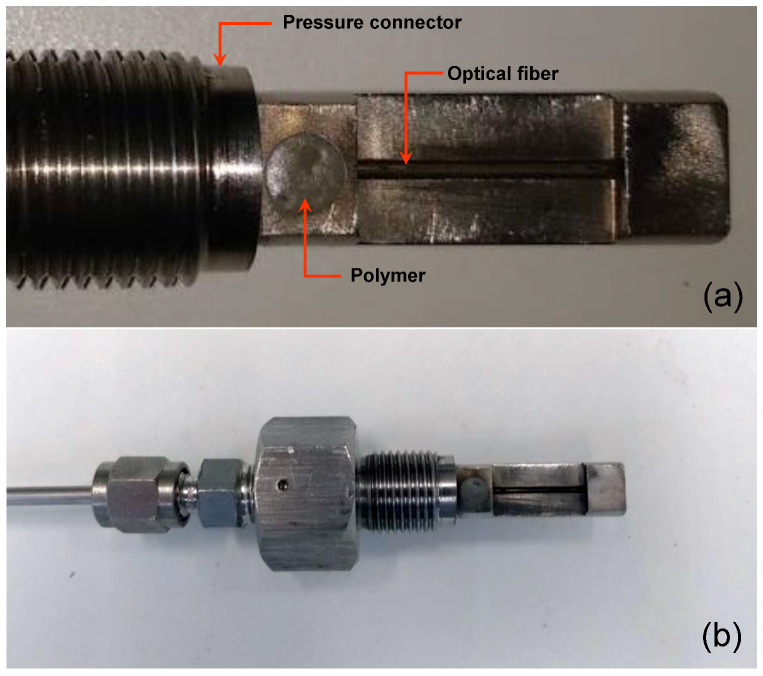
(**a**) Hole of a pressure connector filled with the metallic polymer and (**b**) the pressure connector using the polymer as a sealing technique.

**Figure 4 sensors-21-06531-f004:**
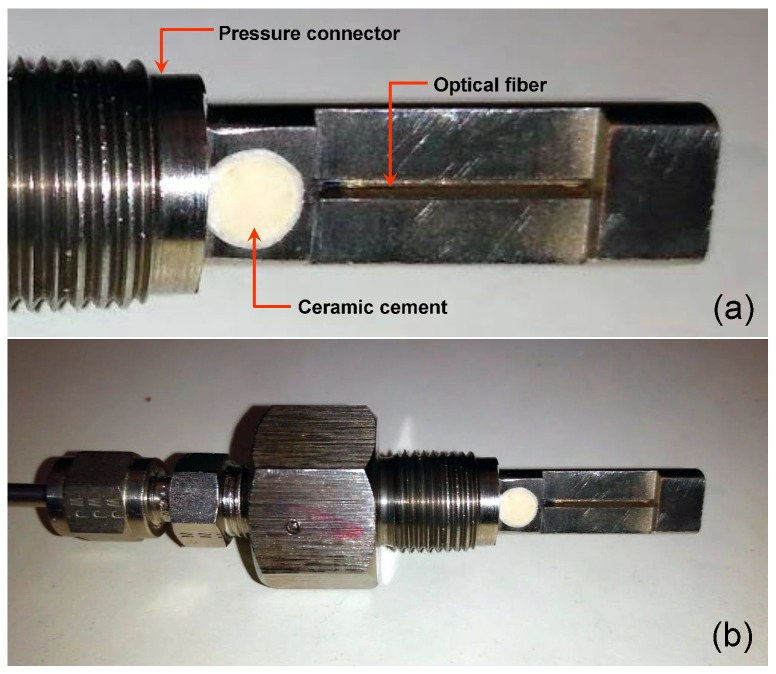
(**a**) Hole of a pressure connector filled with the ceramic cement and (**b**) the pressure connector using the ceramic cement as a sealing technique.

**Figure 5 sensors-21-06531-f005:**
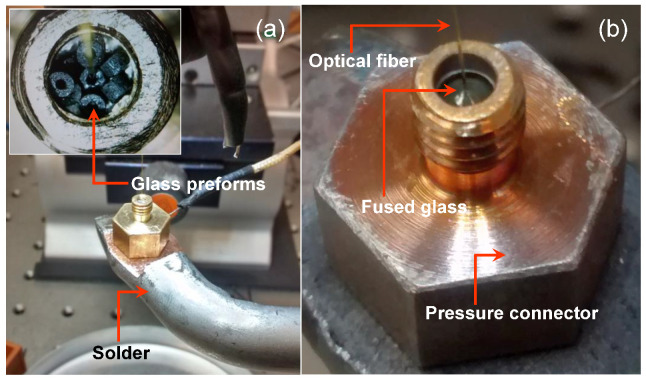
(**a**) Hole of a pressure connector filled with the glass preforms. An iron solder is used to melt the glass preforms and (**b**) a view of the melted glass sealing the optical fiber.

**Figure 6 sensors-21-06531-f006:**
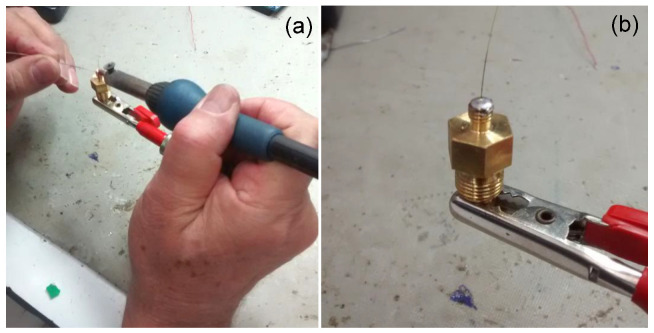
(**a**) Soldering the gold metalized fiber in the pressure connector and (**b**) the pressure connector sealed with HMP and exhibiting the optical fiber.

**Figure 7 sensors-21-06531-f007:**
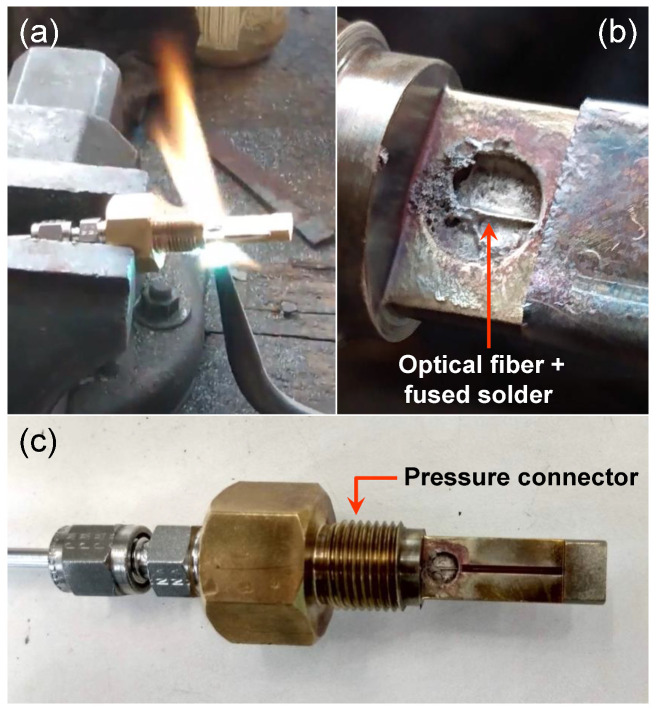
(**a**) the process to solder the gold metalized fiber that contains the FBG in a pressure connector using the SSQ-6 solder. (**b**) the result of the soldering process. (**c**) the entire pressure connector (3/8-24 UNF (M10x1.0).

**Figure 8 sensors-21-06531-f008:**
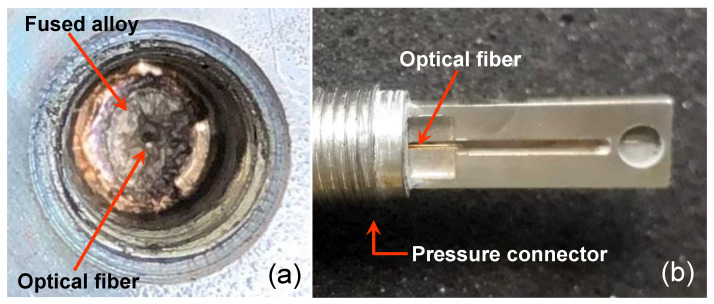
(**a**) Brazing sealing result in the internal part of the pressure connector and (**b**) the external part of pressure connector exhibiting the optical fiber.

**Figure 9 sensors-21-06531-f009:**
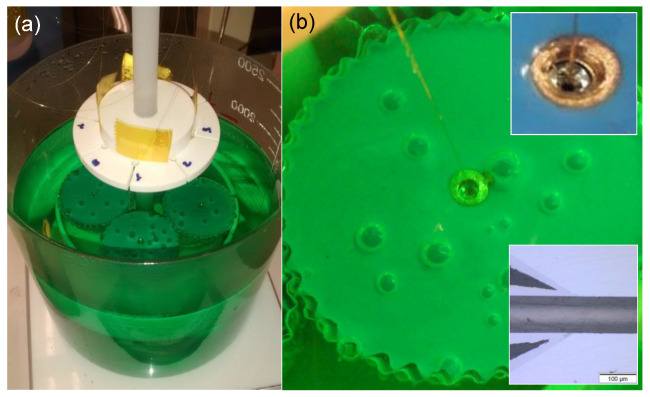
(**a**) Chemical nickel/phosphorous bath during a fiber/ferrule metallization process and (**b**) fiber ferrule sealed in the nickel ferrule after the metallization process.

**Figure 10 sensors-21-06531-f010:**
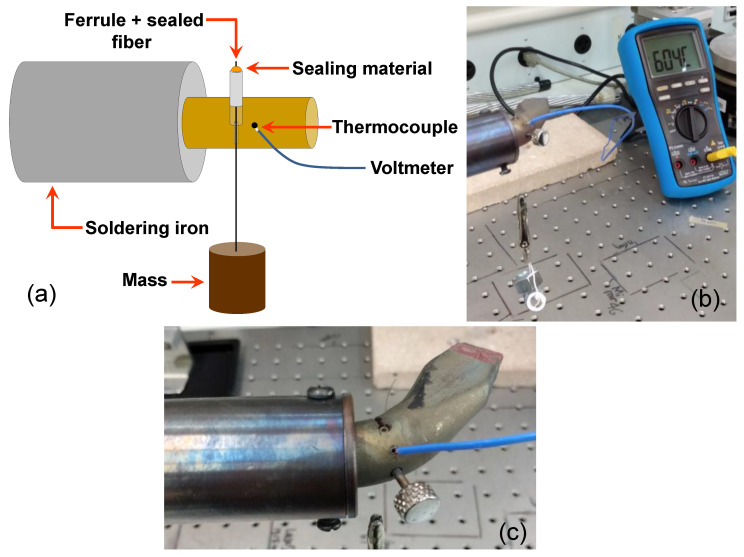
(**a**) Schematic of the setup utilized to test the endurance of sealing, (**b**) photo of the real setup and (**c**) zoom of the ferrule with the fiber sealed in the heating region of the soldering iron.

**Figure 11 sensors-21-06531-f011:**
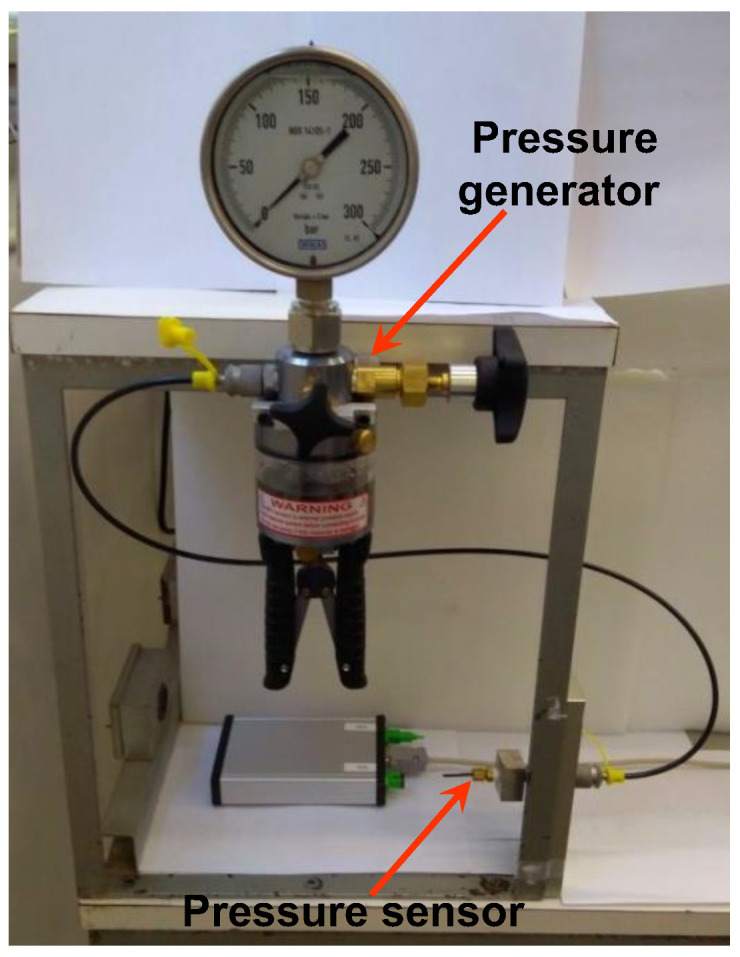
Static pressure generator used in the leakage test.

**Figure 12 sensors-21-06531-f012:**
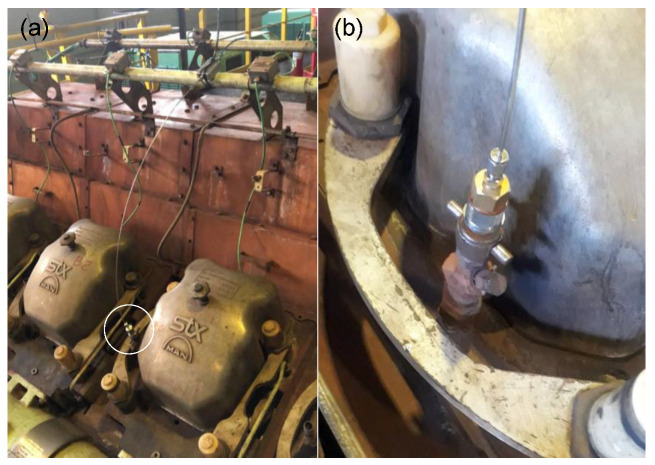
(**a**,**b**) Sensor sealed with polymer in the engine pressure monitoring point.

**Figure 13 sensors-21-06531-f013:**
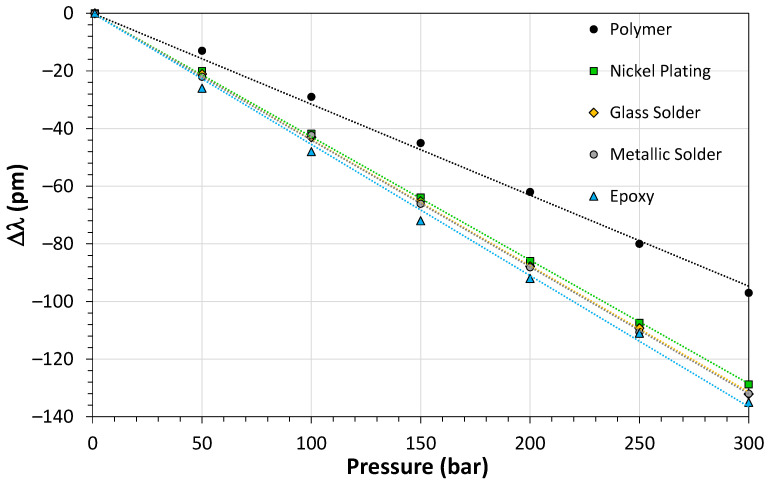
Results for the sensors that were approved in the leakage test.

**Figure 14 sensors-21-06531-f014:**
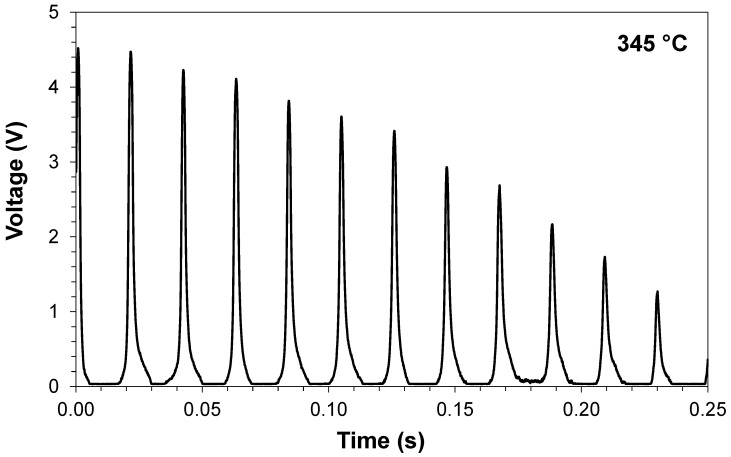
Temporal response of pressure sensor using nickel-plating sealing in the moment that the sealing degradation started.

**Table 1 sensors-21-06531-t001:** Results for endurance, leakage and field-testing.

Material/Sample	Test 1	Test 2	Test 3 ^A^
Epoxy 01	410 °C	>300 bar	5 min
Epoxy 02	382 °C	>300 bar	-
Epoxy 03	345 °C	>300 bar	-
Polymer 01	>585 °C	>300 bar	>120 min ^B^
Polymer 02	>618 °C	>300 bar	>120 min ^B^
Polymer 03	>618 °C	>300 bar	-
Cement 01	>622 °C	<50 bar	-
Cement 02	>619 °C	<50 bar	-
Cement 03	>606 °C	<50 bar	-
Glass solder 01	309 °C	>300 bar	-
Glass solder 02	334 °C	>300 bar	-
Glass solder 03	312 °C	>300 bar	-
Metallic solder 01	302 °C	>300 bar	30 min
Metallic solder 02	282 °C	>300 bar	-
Metallic solder 03	415 °C	>300 bar	-
Nickel plating 01	350 °C	>300 bar	90 min
Nickel plating 02	350 °C	>300 bar	30 min
Nickel plating 03	460 °C	>300 bar	-

^A^ The conditions for Test 3 were pressure +/− 180 bar and average temperature 350 °C. ^B^ Removed from the engine by the operator.

**Table 2 sensors-21-06531-t002:** Comparison of materials/techniques.

Material	Maximum Temperature ^D^	Complexity and Cost ^E^
Epoxy	350 °C	Low
Polymer	500 °C	Low
Cement	871 °C	Low
Glass solder	380 °C	Medium
Metallic solder 1	270 °C	Low
Metallic solder 2	565 °C	Medium
Metallic solder 3	900 °C	High
Nickel plating	900 °C ^F^	Medium

^D^ According the datasheet. ^E^ Material and equipment included. ^F^ [28].

## Data Availability

Data are contained within the article.

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
