# Peer review of "Techniques and Materials for Optical Fiber Sensors Sealing in Dynamic Environments with High Pressure and High Temperature"

_sensors, 2021, doi:10.3390/s21196531_

Round 1

Reviewer 1 Report

Very nice work.  Thorough testing and evaluation of diverse materials for hermetic sealing of optical fibers.  An area overlooked over the years.  Your findings will be of help to other researchers and engineers active in the fields of fiber sensors and real-life applications in harsh environments.

Reviewer 2 Report

This paper calls "Techniques and Materials for Optical Fiber Sensors Sealing in Dynamic Environments with High Pressure and High Temperature" and concerned of experiense of building fiber optic system operating at high pressures and high temperatures. The advantage of this work is quality of presentation - detailed experience of working with different materials as epoxy, polymer composity, cement, glass solder, metal solder. nickel-plating sealing. In my opinion, such experience will be usefull for people who relate with high temperature. pressure sensors. It is a pity that the authors do not consider metal coated optical fibers (producted using freezing technique during optical fiber drawing process).

The main question is motivation for work. How do authors want to use such a measurement system? If it is a part of measurement system authors must say the principle of operation of such a system (FBG, DTS. Brillouin, etc.). Depending from principle of operation measurement systems different ways of realization of high pressure/temperature fiber input will be needed.

Reviewer 3 Report

This paper presents the authors studied various techniques and sealing materials for optical fiber sensors used in dynamic, harsh environments with high-pressure temperatures. The paper is indeed interest of overcome the problems of fragility for harsh environmental sensing, and associated fiber packaging. The paper is well written, and comparison various materials/techniques.

Minor concerns are;

  1. Not sufficient information on, why metallic polymer is a better material for optical fiber sealing.
  2. Is metallic polymer is better for all the fiber types?, such as silica, polymer, PCF, coreless, etc.
  3. Need to address the possibilities of sealing materials for low-temperature applications of Polymer fibers, and high-temperature applications, >1000C of Saphire fibers.
  4. How the sensing sensitivity will affect after material sealing the fiber.
  5. The Conclusion section needs to be more precise with in detail.
